# Intracellular Penetration and Effects of Antibiotics on *Staphylococcus aureus* Inside Human Neutrophils: A Comprehensive Review

**DOI:** 10.3390/antibiotics8020054

**Published:** 2019-05-04

**Authors:** Suzanne Bongers, Pien Hellebrekers, Luke P.H. Leenen, Leo Koenderman, Falco Hietbrink

**Affiliations:** 1Department of Surgery, University Medical Center Utrecht, 3508 GA Utrecht, The Netherlands; shbongers@gmail.com (S.B.); hellebrekers.p@gmail.com (P.H.); l.p.h.leenen@umcutrecht.nl (L.P.H.L.); 2Laboratory of Translational Immunology, University Medical Center Utrecht, 3508 GA Utrecht, The Netherlands; l.Koenderman@umcutrecht.nl; 3Department of Pulmonology, University Medical Center Utrecht, 3508 GA Utrecht, The Netherlands

**Keywords:** neutrophils, polymorphonuclear leucocyte (PMN), intracellular antibiotics, Staphylococcus aureus, intracellular pathogen

## Abstract

Neutrophils are important assets in defense against invading bacteria like staphylococci. However, (dysfunctioning) neutrophils can also serve as reservoir for pathogens that are able to survive inside the cellular environment. *Staphylococcus aureus* is a notorious facultative intracellular pathogen. Most vulnerable for neutrophil dysfunction and intracellular infection are immune-deficient patients or, as has recently been described, severely injured patients. These dysfunctional neutrophils can become hide-out spots or “Trojan horses” for *S. aureus*. This location offers protection to bacteria from most antibiotics and allows transportation of bacteria throughout the body inside moving neutrophils. When neutrophils die, these bacteria are released at different locations. In this review, we therefore focus on the capacity of several groups of antibiotics to enter human neutrophils, kill intracellular *S*. *aureus* and affect neutrophil function. We provide an overview of intracellular capacity of available antibiotics to aid in clinical decision making. In conclusion, quinolones, rifamycins and sulfamethoxazole-trimethoprim seem very effective against intracellular *S. aureus* in human neutrophils. Oxazolidinones, macrolides and lincosamides also exert intracellular antibiotic activity. Despite that the reviewed data are predominantly of in vitro origin, these findings should be taken into account when intracellular infection is suspected, as can be the case in severely injured patients.

## 1. Introduction

Neutrophils are the first line of defense against invading bacteria, preventing and clearing infection continuously [1]. Neutrophils are recruited to the site of infection where they recognize bacteria using Fc receptors, after which the cell membrane remodels around the receptor-ligand complex to form a phagosome [2,3]. Inside cells, the phagosome fuses with secretory vesicles and granules to form the phagolysosome, which include bactericidal enzymes, vacuolar ATPases and the NADPH oxidase complex. Inside the phagolysosome, the acidic milieu, caused by proteases, reactive oxygen species (ROS), antimicrobial peptides and other processes cause intraluminal degradation of the bacteria [4]. Additionally, neutrophils play an important role in the salvage of tissue damage and tissue repair after infection or in case of trauma [5,6,7].

Dysfunctional neutrophils are not only prominent in patients with congenital neutrophil dysfunction but are also seen in several acute inflammatory conditions and severely injured patients [8,9]. In severely injured patients, both bacterial killing capacity and, to a lesser extent, phagocytosis are diminished [10]. This results in decreased bacterial clearing and thereby increased susceptibility to infection in an already vulnerable patient group [11]. In the past few decades, a steady decline is seen in early death rates after trauma [12]. Infectious complications are an increasingly important cause of morbidity, prolonged hospital stay and decreased functional recovery [13]. Common infections in trauma patients encompass pneumonia, bacteremia, surgical site infection (SSI) and fracture-related infections (FRI) [14].

*Staphylococcus aureus* (*S. aureus*) is a common causative pathogen in SSI or FRI [15,16,17]. *S. aureus* is notoriously known for its biofilm formation, especially in infections involving medical implants [18]. *S. aureus* has several other defense mechanisms, one of which is the ability to survive inside the phagosome of host cells [19]. Although this covers mostly non-professional phagocytes (e.g., endothelial cells, epithelial cells, osteoblasts), it has also been described in diseases with neutrophilic dysfunction or overwhelming bacterial numbers [20,21], both of which may be the case in severely injured patients. 

Neutrophils are shown to be able to transport living intracellular pathogens, causing and even promoting distant infections in sandflies [22]. Recently, it has also been shown that *S. aureus* is able to survive and proliferate inside neutrophils of LPS-challenged human volunteers [9]. Thwaites and Gant also make a compelling case for the metastasis of *S. aureus* infections in humans by survival of the pathogen inside neutrophils and the detrimental role of the neutrophil in *S. aureus*-bacteremia [23].

To achieve protection against infections in vulnerable patients such as the severely injured, prophylactic antibiotics are frequently administered, especially in the intensive care unit (ICU) setting [24,25]. However, a major part of the most common empirical antibiotics are not able to target intracellular pathogens [26]. To our knowledge, an overview of the intracellular effect of antibiotics on *S. aureus* residing in neutrophils is missing. Therefore, we conducted a literature review to summarize the properties of commonly used antibiotics regarding their ability to enter neutrophils, the intra-cellular bactericidal or bacteriostatic effect on *S. aureus* and their effects on neutrophil functions regarding intracellular killing.

## 2. Results

The conducted search and subsequent in- and exclusion resulted in a total of 110 articles included in this review. The gross majority of the articles only provided in vitro data (*n* = 98), but some also showed in vivo/ex vivo data (*n* = 12). For legibility, the data discussed in this review are of in vitro origin, unless stated otherwise. 

A summary of the extracted data per antibiotic class is shown in Table 1. In Table 1, the results are arranged in a specific order: Degree of intracellular penetration, effect on neutrophil function (e.g., phagocytosis, reactive oxygen species (ROS) production, antibacterial capacity), degree of intracellular effect of the antibiotic on *S. aureus* and the type of this antibiotic effect (static or cidal). This order matches the order of discussion of these subjects for each different antibiotic in the text below. In Table 1, the intracellular penetration is expressed by the cellular/extracellular (C/E) ratio of the drug. C/E values are calculated by dividing the intracellular concentration of the drug by the extracellular concentration. Antibiotics described in the results section are sorted based on their mechanism of action and class.

### 2.1. Protein Synthesis Inhibitors

#### 2.1.1. Aminoglycosides

Generally, aminoglycosides (e.g., gentamicin) do not accumulate in neutrophils. C/E ratios mostly stay below 1 [28,29,38,39,40]. It is believed that aminoglycosides penetrate poorly because of their high polarity and hydrophilic characteristics [28,41,42,43]. Intracellular aminoglycosides seem to relatively accumulate in lysosomes, while the low pH in lysosomes inhibits their antibacterial function [44].

Aminoglycosides at clinically relevant concentrations do not seem to affect properties that are important for the intrinsic killing capacity of neutrophils (like ROS production in activated neutrophils) [30,45]. However, at very high doses (above achievable serum concentrations), aminoglycosides seem to have a direct toxic effect on neutrophils and thereby inhibit neutrophil function [31,32].

Most frequently used aminoglycosides (i.e., gentamicin) have no convincing effect on intracellular *S. aureus* at clinically relevant extracellular doses (≤5 mg/L) [28,31,33,34,42,43,44]. Gentamicin, however, remains in an active form inside neutrophils, indicating no intracellular inactivation [28]. In higher extracellular concentrations (from 5 to 25 mg/L), gentamicin and streptomycin show some reduction in viable intracellular bacteria [35,40]. This is probably due to intracellular concentrations above the minimum bactericidal concentration (MBC), despite minimal cellular penetration [40]. This effect on killing seems to be due to the direct antibacterial effect of the antibiotics [35]. In contrast to other aminoglycosides, tobramycin and arbekacin seem to have a very pronounced bactericidal effect against intracellular *S. aureus* [36,37]. This observed effect was partly but not completely due to overestimation of the amount of killed intracellular bacteria [36]. An exact distinction between the direct effect of tobramycin and the synergy with neutrophils in the process of killing could not be made [36]. An explanation for the differences between different aminoglycosides has not been found.

#### 2.1.2. Tetracyclines

Tetracycline seems to moderately penetrate and accumulate inside neutrophils, reaching C/E ratios of 1.8–7.1 [46,47,48]. Uptake of tetracycline is relatively slow—80% of its final intracellular concentration is reached after 40 min [47]. Accumulation is more extensive with doxycycline (C/E ratio 7.5). Other less known tetracyclines (tigecycline and minocycline) reach even higher C/E ratios, up to 64 [49]. In vivo C/E ratios might be lower due to serum protein binding of these antibiotics [48,49]. Uptake of tetracyclines into neutrophils is saturable and seems to be through active organic cation transport with a relatively low affinity [50]. It has been suggested that there is a lack of intracellular binding, resulting in high intracellular bioavailability [48].

Tetracyclines, like tigecycline, seem to stimulate ROS production of activated neutrophils at clinically relevant concentrations, reaching a plateau at 5–10 mg/L, as reported by Cockeran et al. [50]. Tetracyclines seem to have a calcium ionophore function, where tigecycline also seems to scavenge ROS. These properties counteract each other and cause induction of the before mentioned plateau phase [50]. Contrarily, Naess et al. report no effect of tigecycline on both phagocytosis and oxidative burst formation at doses 0.1–100 mg/L [51]. These opposing results might be due to methodological differences. Naess et al. used *S. aureus* as a stimulus to activate neutrophils, whereas Cockeran et al. used the chemotactic peptide N-Formylmethionyl-leucyl-phenylalanine (fMLP). Altogether, there is no consensus on the effect of tetracyclines on neutrophil functions.

Regarding the intracellular antibacterial effects, only Ong et al. investigated the intracellular activity of the tetracycline antibiotic tigecycline and demonstrated bacteriostatic activity against *S. aureus* at 1 mg/L [49].

#### 2.1.3. Macrolides

Uptake of macrolides into neutrophils can be divided into two distinct groups. First, there is erythromycin (C/E between 4.4 and 13.3) [29,38,41,52,89,90], clarithromycin (C/E of 9.2) [52], roxithromycin (C/E between 14 and 34) [41,53,90], dirithromycin (C/E of 16.3) [54] and josamycin (C/E of 21.4) [55]. These macrolides penetrate neutrophils moderately well, generally reaching maximal C/E values after 10 to 90 min of incubation [52,56,57,58,89]. Uptake of this group of macrolides might be slightly inhibited by environmental factors (serum proteins, low pH, low temperatures) [41,56] or the state of the neutrophil (activated, phagocytosis) [40,53,56,59,60,61]. Release from neutrophils is fast upon washing, indicating weak intracellular binding [29,38,58,89]. Second, there are azithromycin and telithromycin. These drugs reach very high C/E values of >100 generally, with even higher maxima after relatively long incubation periods of 40 min up to 24 h [57,58,62,63]. Release of azithromycin and telithromycin is slow, which indicates extensive trapping inside cells [58]. Phagocytosis stimulates release of azithromycin [57]. In vivo data also show very high C/E ratios for azithromycin and clarithromycin at clinically relevant concentrations, yet to a somewhat lesser extent [64,65,66]. A good explanation for such high C/E values is a close association with or binding to cellular components [67]. Overall intracellular uptake kinetics of azithromycin are comparable between healthy volunteers and immunocompromised patients [68,69]. In vivo data regarding uptake are in line with in vitro data [70]. Uptake of macrolides seems to be at least partly active and energy-requiring, mediated by the nucleoside transporter [29,47,53,63,89] and the cellular sodium/calcium exchanger on the neutrophil’s cell membrane [63,71]. Despite this active mechanism, the Km (Michaelis kinetic constant) for this nucleoside transporter is higher than the therapeutic concentrations reached in vivo. As a result, in vivo erythromycin most likely follows the linearly increasing part of the saturation curve [89]. In addition, passive uptake also plays a significant role [47,55,56,57,72]. Intracellularly, macrolides are taken up into granules for 33–73% [41,58,61], by diffusion, lipophilicity and ion trapping, causing high intracellular concentrations [41,57,62,64,73]. Besides ion trapping, a specific transportation protein is thought to be involved in the uptake of macrolides into the granule compartment of neutrophils [58].

At therapeutic levels, macrolides do not show clear effects on neutrophil function [57,74,75], yet some evidence suggests a slight induction of ROS production [54,74,76,77,78]. However, at higher doses or longer exposure times, an inhibitory effect on ROS production becomes prominent [30,54,74,76,77,78,79,80]. No consensus is reached on the underlying mechanism of the inhibitory effect. Different theories have been proposed: Reduced expression of beta-2-integrins [81], inhibition of the phospholipase D-phosphatidic acid phosphohydrolase (PAH) interaction with protein kinase C activity and altered construction of the NADPH oxidase enzyme [54,78,82], membrane stabilization [54], elevation of cyclic adenosine monophosphate (cAMP) levels and higher protein kinase A (PKA) activity [32,80]. Erythromycin is also seen to induce LDH release and neutrophil death even at clinically relevant concentrations (1–10 mg/L) [32]. In spite the inhibitory effect of telithromycin and roxithromycin on the oxidative metabolism of neutrophils, these antibiotics do not impair the antibacterial neutrophil function [60].

In both in vitro and ex vivo data, macrolides (erythromycin, roxithromycin, azithromycin and clarithromycin) show bacteriostatic but not bactericidal intracellular effects at therapeutically relevant extracellular concentrations [36,40,52,62,64,75,83,84,90]. Others contradict the intracellular efficacy [40,85]. These results may be explained by the lack of sensitivity of the applied method for analyzing the intracellular bacteriostatic activity [90], short incubation times and usage of below therapeutic levels of antibiotics [40]. Bacteriostatic effects for macrolides are less pronounced in neutrophils with defective oxidative killing mechanisms. This indicates a synergistic effect with the intrinsic killing mechanisms of neutrophils for this type of antibiotic [62,83,84,86,87,90]. Contrarily, roxithromycin selectively restores functions of defective neutrophils, as low doses increase phagocytosis and intracellular killing of *S. aureus* [88].

#### 2.1.4. Lincosamides

Clindamycin uptake is good, reaching C/E ratios ranging from 8 to 43.4 within minutes [29,38,43,91,92,93]. This is temperature-dependent, declining at temperatures below 37 °C [61,91]. Uptake of lincomycin is remarkably less with maximum measured C/E ratios of 3 [29,38,93]. High pH and/or phagocytosis increase uptake of both clindamycin and lincomycin, which partly explains the variation between experiments that measured C/E ratios for clindamycin [30,38,40,59,91,94,95]. In vivo binding to serum proteins might lower the intracellular uptake [36]. A saturable uptake mechanism has been observed, suggesting an active mechanism of uptake [29,91]. The nucleoside transporter seems to be a reasonable option because clindamycin possesses a nucleoside-like structure [91], and adenosine uptake is inhibited by clindamycin [38,59]. Intracellularly, clindamycin is thought to accumulate in the lysosomal compartment [91] or cytoplasm [61]. Efflux of clindamycin out of neutrophils is varying in different experiments. Some report that efflux of clindamycin is rapid, indicating no tight intracellular associations [29]. Others state that clindamycin slowly eliminates from neutrophils during 24 h [43].

At serum concentrations of 1.0 mg/L, clindamycin slightly enhances neutrophil phagocytosis in healthy volunteers [96]. However, contradictory results are found regarding the effect on the oxidative metabolism of neutrophils (induction or depression) [30,96]. Postulated inhibitory effects could be due to extra binding of displaced adenosine to its A2 receptors [30] or a decrease in glycerol production and subsequently less ROS production [78]. Effects on oxidative metabolisms do not seem to affect the bacterial killing capacity [96]. This supports the independence of the oxidative killing mechanisms regarding the antibacterial effect of the drug [45,97].

Impressive reduction (up to 96%) of intracellular bacteria occurs in the presence of clindamycin at concentrations resembling clinically achievable serum concentrations [34,36,42,83,91,92]. In vivo effects after oral dosing with clindamycin are comparable to in vitro results [97]. This effect is partly due to synergism between the oxidative killing mechanisms of neutrophils and the antibiotic [83,98]. The direct effect of clindamycin is believed to be bacteriostatic [36,83,94]. Despite these positive results, a lack of effect on intracellular killing for both clindamycin and lincomycin at extracellular concentrations up to 20 mg/L is also reported more than once [31,40,94,99]. A relatively short incubation time used in these studies might cause this observed lack of effect due to the relatively slow onset of intracellular action of lincosamides [34].

#### 2.1.5. Oxazolidinones

Radezolid reaches a C/E ratio of roughly 11. The uptake is rapid (<30 min) and non-saturable. Less accumulation is seen at pH <6, at lower temperatures and in the presence of serum protein. This argues for a passive diffusion mechanism of uptake and subsequent ion-trapping [100]. Linezolid accumulates poorly, proposedly because of its charge and lesser lipophilicity, resulting in impaired diffusion into the cell. Linezolid reaches a lower C/E ratio of around 1, but this is only tested in THP-1 cells [100]. After uptake, radezolid accumulates for 42% in lysosomes and for 58% in the cytosol. No association with cellular components, like mitochondria, is seen. These characteristics were only studied in macrophages and not in neutrophils, in which these characteristics might differ [100].

Despite its poor accumulation inside phagocytes, linezolid has good intracellular activity in neutrophils. Linezolid showed a marked decrease in survival—only 1.65% intracellular MRSA survival (compared to control, 100% survival) at concentrations of 18 mg/L after 24 h [37]. 

### 2.2. Interference with the Cell Wall (Beta-Lactams)

#### 2.2.1. Penicillins

Penicillin antibiotics penetrate neutrophils poorly, with C/E ratios <1, due to hydrophilicity and in vivo serum protein binding [29,38,40,42,43,46,55,61,93,101]. Cloxacillin is the exception, with C/E values of 4.7 [95]. It is unclear whether activation of the neutrophil/phagocytosis affects uptake—some studies indicate it does affect uptake [31,95], while some find no effect on uptake [40,59].

Benzylpenicillin, dicloxacillin and oxacillin exert no effect on neutrophil properties like ROS production [30,45,102]. In contrast, ampicillin seems to have a small inhibiting effect on myeloperoxidase (MPO) activity and phagocytosis [103]. No subsequent effect on bacterial killing is reported [32]. Ampicillin and dicloxacillin have no effect on apoptosis and viability of neutrophils, even at high concentrations [32,102,104].

Pre-treatment of bacteria with benzylpenicillin at concentrations below minimum inhibitory concentration (MIC) and in the presence of serum causes a noteworthy reduction in intracellular colonies [33,105,106]. The pro-killing effect of pre-treatment with penicillin is a result of direct effects of the antibiotic on the bacterial cell wall and mitotic activity. This renders bacteria more susceptible to neutrophil killing [106]. In addition, exposing neutrophils to sub-MIC concentrations of penicillin after phagocytosis causes enhanced bacterial killing despite poor uptake [31,98].

The intracellular antibacterial effects of penicillins are controversial. Some authors state that penicillin antibiotics at clinically relevant concentrations (and higher) have near to no effect on killing of intracellular bacteria [33,34,35,40,42,43,107,108]. However, other studies do show a concentration-dependent antibacterial effect on intracellular bacteria with various penicillin antibiotics (benzylpenicillin, amoxicillin, dicloxacillin) at clinically relevant extracellular concentrations [36,83,102,104,109,110]. Dicloxacillin seems to have both bacteriostatic and bactericidal effects at higher concentrations [36,102]. Penicillins interact with the oxidative killing system of neutrophils to exert their alleged intracellular effects [83]. Roder et al. suggested that the intracellular effects seen with penicillins might be partly caused by antibiotic carryover during the lysis procedure [36].

#### 2.2.2. Glycopeptides

Glycopeptides reach C/E ratios of 4–4.5 after 20 min of incubation, with C/E ratios of 7.8 after neutrophil activation [95]. Others report C/E ratios up to 52.3 for teicoplanin [31,92]. Glycopeptides are thought to bind avidly to neutrophil associated proteins [111], which might reduce their intracellular actions [92]. Intracellular teicoplanin does not enter (phago)lysosomes, further hampering antibacterial effects [92].

The glycopeptides do not interfere with neutrophil functions like chemotaxis, adherence, phagocytosis, NADPH oxidase function and ROS production [45,112].

Pre-treatment of *S. aureus* with low concentrations of vancomycin or teicoplanin in the presence of serum enhances susceptibility to killing by neutrophils [106,112]. This might be due to modification of bacterial characteristics by modification of opsonization or interaction between the antibiotic and the bacterial inoculum in vitro before ingestion [31].

Several studies report that glycopeptides show no intracellular bioactivity against *S. aureus* [31,37,113]. Others, however, do see a significant dose- and time-dependent inhibition or even bactericidal effects on intracellular *S. aureus* at the upper limits of clinically achievable extracellular concentrations [36,111]. This effect seems slightly more pronounced for teicoplanin than vancomycin [111].

#### 2.2.3. Carbapenems

Imipenem is rapidly accumulated inside neutrophils, but only reaches C/E ratios between 1 and 5 [53,61,95]. Surprisingly, right after this fast accumulation, a rapid and steady decline is seen over time [53,61]. The cause of this phenomenon is unknown. It might be due to binding to the cellular membrane and subsequent rapid dissociation, alterations to the drug molecule or cellular metabolism of the drug [53]. Imipenem seems to accumulate outside the granule fraction of neutrophils, since it noticeably accumulates more inside cytoplasts (without granule fraction) than in normal neutrophils [61].

Carbapenems do not seem to exert effect on the properties and killing abilities of neutrophils [114].

Meropenem shows a small but significant reduction in intracellular survival of *S. aureus* at 8 times MIC and at 6 h of incubation. Killing is relatively slow. On the contrary, imipenem shows no effect on intracellular survival [114].

#### 2.2.4. Cephalosporins

Cephalosporins are hydrophilic drugs, and are therefore unable to penetrate the cell membrane [42,43]. This results in C/E values <1 [29,38,53,93,115]. Sequestration in phagocytic vacuoles might increase uptake of cephalosporins during phagocytosis in active neutrophils [115].

Cephalosporins like cefazolin have no effect on apoptosis of neutrophils at therapeutically relevant concentrations [32]. Both in vitro and in vivo data show a lack of effect on neutrophil peroxide production, intracellular calcium changes, degranulation and ROS production [115,116]. In vivo data did not show restoration of phagocytosis and bactericidal activity in patients with decreased neutrophil functions [88]. On the contrary, patients with defective neutrophils seem to have increased neutrophil MPO activity after cefaclor treatment and show better clinical recovery compared to therapy with other (non-cephalosporin) antibiotics [103].

Incubation of *S. aureus* with sub-MIC concentrations of cephalosporins (up to 2 mg/L) causes formation of enlarged forms of *S. aureus* more susceptible to phagocytosis, increased phagocytosis and increased killing efficacy of neutrophils. This is related to cell wall modifications of bacteria, as seen in penicillins and glycopeptides [98,117,118].

At therapeutically relevant concentrations, cephalosporins are generally ineffective against intracellular *S. aureus* [35,42,43,115,118]. This is mostly explained by the lack of cellular penetration and further enhanced by the relatively slow intracellular bacterial growth reducing the effect of glycopeptides [35]. Notably, cefodizime shows a modest but significant increase in phagocytosis and intracellular killing at high therapeutically concentrations (100 mg/L) [116].

### 2.3. Nucleic Acid Inhibitors

#### 2.3.1. Sulfonamides

Sulfamethoxazole reaches C/E ratios 1.7–3.6 in neutrophils within 10 min [46,119]. Despite the slightly acidic characteristics of sulfamethoxazole, it still moderately accumulates in the acidic granulocytes and its (phago)lysosome [42,46,120]. Phagocytosis enhances uptake of these antibiotics [120]. The degree of uptake of different sulfonamide antibiotics is as follows (best to worst): Sulfanilamide > sulfadiazine > sulfamerazine > sulfamethoxazole. This is not in line with the degree of lipid solubility of the different drugs, pointing towards involvement of an active mechanism of uptake. Intracellularly, sulfanilamide distributes within neutrophils: 35–40% in the cytosol, 35–40% microsomal, 10–20% association with membranes and 10% association with mitochondria or the nucleus [120]. It is unknown whether the same distribution profile applies to other sulfonamides.

Since sulfamethoxazole is frequently used in combination with DHFR inhibitor trimethoprim, intracellular effects of this combination on killing of *S. aureus* will be discussed later on in this review (see subheading “Combinations”).

#### 2.3.2. DHFR Inhibitors

Trimethoprim shows rapid uptake with C/E ratios 3–21 after 10 min [42,43,46,53,93,120,121]. Remarkably, uptake increases under lower than physiological temperatures (25 °C) [53,61]. This phenomenon might be explained by a rapid efflux due to absence of binding to cellular components at 37 °C [61,121]. Uptake and efflux are thought to be passive processes [121]. Trimethoprim is weakly basic, which favors accumulation into relatively acidic neutrophils and their granules [46,121]. Brodimoprim shows efficient penetration, reaching a C/E ratio of 74.4, as a result of high lipid solubility [120,121]. The intracellular distribution profile for trimethoprim is similar as described above for sulfanilamide [120]. Trimethoprim accumulates better in neutrophils than in cytoplasts, due to its weakly basic properties and the positive effect of the presence of granules [61].

Trimethoprim seems to have a dose-dependent inhibitory effect on superoxide production in neutrophils due to inhibition of PAH pathways [30,121]. This does not influence neutrophil properties like phagocytosis and killing [121].

Effectiveness of intracellular killing of trimethoprim will be discussed in combination with sulfamethoxazole (see subheading “Combinations”).

#### 2.3.3. Quinolones

Quinolones reach C/E ratios of 2.2–10.9 within 30 min in both in vitro and in vivo settings [93,95,105,122,123,124,125,126,127,128,129]. After one oral dose, intracellular concentrations (±9 mg/L) and area under the curve (AUC) values reach and remain higher compared to those of plasma [129]. Active phagocytosis does not alter the quantity of uptake of quinolones [95,105,123,126,130,131]. In the case of ofloxacin optical isomerism influences uptake [125], rapid elution suggests a lack of binding of ciprofloxacin to cellular components [31,105]. Penetration of quinolones seems to be partially due to a passive mechanism since it is pH- and temperature-dependent and does not rely on viability of neutrophils [105,123,126]. Additionally, active transportation expressing Michaelis–Menten kinetics is observed for ciprofloxacin [130]. Metabolic inhibitors, inhibitors of PKC and MAP kinase [126,130] and the addition of amino acids [131] inhibit uptake. An exception is gatifloxacin, of which transport is not influenced by these inhibitors [127]. Uptake of these drugs is less in promyelocytic HL-60 cells, indicating the need for maturation and expression of certain neutrophil proteins in order to penetrate the cell [130]. It is suggested that uptake in resting neutrophils is facilitated by a nucleobase transportation mechanism (low affinity), and uptake in active neutrophils is facilitated by the amino acid transportation system (high affinity) [131]. Preference for one of the described uptake mechanisms may differ between quinolones, depending on their molecular structure.

Quinolones generally do not affect neutrophil phagocytosis, ROS production, intracellular bacterial killing or viability at clinically relevant doses and higher [45,88,104,105,107].

Quinolones are found to be biologically active inside neutrophils [31,105]. Among others, ciprofloxacin and ofloxacin significantly reduce intracellular viable bacterial counts in a concentration and time-dependent manner at (sub)therapeutic concentrations as low as 0.4 mg/L [31,84,104,105,125,126,132]. Intracellular ciprofloxacin is bacteriostatic and bactericidal at extracellular concentrations of 0.5 mg/L and 2 mg/L, respectively [36]. Other data show that ciprofloxacins’ and enoxacins’ intracellular activity is poor at MIC [31], but increases significantly at 4–16 x MIC [104]. In the study of Yanai et al., a lack of effect of ofloxacin in chronic granulomatous disease (CGD) neutrophils is shown at doses 50 mg/L after an incubation period of 1 h. However, they also show a lack of direct activity at high doses against the tested strains of *S. aureus*, which is suggestive of methodological failure [107]. The intracellular effect is remarkably less when CGD neutrophils are used—higher doses (2,5 mg/L) of ciprofloxacin are needed to achieve significant effects. This indicates the need for interaction of quinolones with the oxidative antimicrobial system of neutrophils in order to achieve optimal antibiotic effects [132].

#### 2.3.4. Rifamycins

Rifampin is a zwitterion and lipid soluble, which causes it to accumulate inside neutrophils [29,33,101]. Penetration of rifampin is based on passive mechanisms [29,133]. Rifampin reaches moderate C/E ratios of 2.3–9.8 within 20 min [29,38,39,40,93,94,119,133]. Rifapentine reaches higher C/E values of 4.5–87.6 [92,134]. Activation of neutrophils does not influence uptake of rifampin [36,40,83,94]. In vivo serum protein binding negatively influences penetration into neutrophils [36]. After entering the neutrophil, rifampin is likely to accumulate inside the phagosome [44].

No effects on neutrophil functions, like ROS production and the intrinsic killing mechanisms, were seen with rifampin and rifabutin at clinically relevant concentrations and higher [45,133].

Rifampin and rifapentine are able to strikingly reduce intracellular *S. aureus* in neutrophils in a concentration and time-dependent manner due to bactericidal action at therapeutically relevant extracellular doses, as low as 0.0125 mg/L [33,35,36,37,39,40,42,43,44,52,92,94,101,110,113,119,133,134]. Even complete eradication of intracellular bacteria is seen at concentrations starting at 0.1 mg/L [52], and in in vivo situations after oral rifampin therapy of 300 mg three times daily [108]. This effect of rifampin and rifabutin fades out at extracellular concentration below MIC of 0.004 and 0.001 mg/L, respectively [31]. The same results were seen in CGD neutrophils [34,35,43,44,119]. The intracellular effect mostly seems to be due to direct antimicrobial effects rather than secondary effects on neutrophils and their intrinsic killing mechanism [43,134].

### 2.4. Others

#### 2.4.1. Fosfomycin

Fosfomycin reaches C/E ratios of 1.8 [119]. As fosfomycin is a hydrophilic molecule, and uptake can be hindered by metabolic inhibitors, it is most likely to be actively transported into neutrophils [119].

Fosfomycin enhances phagocytosis and intracellular bactericidal capacities through upregulation of NADPH-dependent ROS production and consecutively enhanced neutrophil extracellular traps (NET) formation [135].

Fosfomycin at clinically relevant doses (8–200 mg/L) restores intracellular killing of defective neutrophils, but does not seem to enhance intracellular killing in healthy neutrophils [119]. However, in another study, fosfomycin (150 mg/L) did show a bactericidal effect, inducing shrinkage and cell wall lysis of *S. aureus* inside healthy neutrophils [113].

#### 2.4.2. Daptomycin

Daptomycin penetrates neutrophils poorly, only reaching a C/E ratio of 0.7 [31,136].

Daptomycin does not affect neutrophil properties like ROS production, phagocytosis and killing at extracellular concentrations up to 200 mg/L [136].

Daptomycin shows no effect on intracellular killing of *S. aureus* at MIC (0.2–0.8 mg/L) [31,136]. However, daptomycin does show a small effect on preventing overgrowth of bacteria inside neutrophils at relatively low concentrations [136].

### 2.5. Combinations

#### 2.5.1. Sulfamethoxazole-Trimethoprim

Uptake kinetics and C/E values for sulfamethoxazole and trimethoprim were discussed previously in this review. 

Sulfamethoxazole-trimethoprim showed no effects on neutrophil properties like catalase activity, oxygen metabolism or killing mechanisms at clinically relevant doses (100/20 mg/L) [46].

Sulfamethoxazole-trimethoprim is highly effective in reducing intracellular *S. aureus* at clinically relevant concentrations in both healthy and CGD neutrophils (i.e., 80/4 or 50/2 mg/L) [37,42,43], killing almost all intracellular bacteria after an incubation period of 24 h [37]. This is due to a direct antibacterial effect [42,43]. These effects make sulfamethoxazole-trimethoprim a preferred antibiotic in the treatment of CGD-related infections [34,46,119].

#### 2.5.2. Amoxicillin-Clavulanic Acid

Uptake kinetics of clavulanic acid alone are not discussed in this review, but data about amoxicillin were discussed previously in this review. 

Only one study conducted by Pascual et al. has been found which evaluates this antibiotic combination [137]. They find that amoxicillin-clavulanic acid induces ROS production and increases intracellular killing of *S. aureus* at high concentrations of 100 mg/L. However, at lower and clinically relevant concentrations, this effect is probably nonexistent. At sub-MIC concentrations, this combination induced phagocytosis of *S. aureus* after pre-incubation with amoxicillin-clavulanic acid. This effect is likely caused by morphological or physiochemical alterations of the bacterial surface, making bacteria more recognizable for neutrophils (described before for amoxicillin) [137].

## 3. Discussion

### 3.1. Findings and Interpretations

Good intracellular effectivity against *S. aureus* inside neutrophils is reserved for certain classes of antibiotics (see Table 1). Based on this review, the most effective classes are the rifamycins, quinolones and the combination sulfamethoxazole-trimethoprim. Bacteriostatic antibiotics, like macrolides and lincosamides, also show good intracellular effectivity. Additionally, oxazolidinones seem promising but still lack sufficient, unambiguous evidence proving their effectivity. Evidence is also short for tetracyclines and fosfomycin. Contradicting data were found regarding penicillins. Other classes like the aminoglycosides, glycopeptides, carbapenems, cephalosporins, individual antibiotic daptomycin and the combination amoxicillin-clavulanic acid do not exhibit convincing effectivity against intracellular *S. aureus*.

Antibiotics do not only exert effects on bacteria but can also influence the intrinsic neutrophil functions (see Table 1). Evidence points towards a dysregulation of neutrophil function after trauma. This could possibly explain the infectious complications seen in these patients [10]. Considering this, it is reasonable to state that stimulation of neutrophil functions like ROS production and boosting the intrinsic killing mechanisms would be beneficial in these patients [138]. On the other hand, heightened ROS production is seen in trauma patients. This can have deleterious effects on bystander tissues when ROS escape the phagosome and neutrophil [10]. In any case, the antibiotic of choice should not inhibit neutrophil killing function even further. Generally, none of the studied antibiotics showed a pronounced effect on neutrophil functions. However, one study on fosfomycin did show some stimulation of functions in defective neutrophils. Studies on tetracyclines prove to be contradictory, showing both stimulation and inhibition of neutrophil capacities. Macrolides are stimulatory at low doses but become inhibitory at higher doses. Other antibiotics generally do not show stimulation nor clear suppression of neutrophil functions. The fact that effects on the oxidative metabolisms do not automatically affect the bacterial killing capacity is noteworthy. Most effects of the studied antibiotics on neutrophil functions like ROS production did not affect neutrophil killing capacities. This implies that non-oxidative killing mechanisms also play an important role in disarming *S. aureus* [96]. Taking all this together, the effects of antibiotics on neutrophil functions do not seem to be the decisive factor for the intracellular effectivity of an antibiotic drug.

A lack of intracellular effectivity can be caused by different factors. First, it can be caused by poor penetration causing insufficient antibiotic concentrations inside neutrophils. For example, drugs that have a low lipophilicity, charged molecules and drugs that are not recognized by transporter proteins in the cellular membrane are less able to penetrate the neutrophil membrane. As a result, the intracellular concentration might not exceed the MIC/MBC of the specific bacterial strain [40,87]. This seems to be the case for the antibiotics with a low C/E ratio. When developing new antibiotic drugs, the ability to penetrate the membrane should be taken into account. An antibiotic that is lipophilic, not charged at physiological pH or an antibiotic recognizable for membrane carriers would be beneficial. The delivery into the cell and the phagolysosome can also be highly improved by using nanocarrier systems [26]. Second, mainly for bacteriostatic agents (i.e., cell wall inhibiting drugs like penicillins), a decreased growth rate of bacteria inside neutrophils negatively influences antibacterial potential [31,35,60,90,109]. Third, tight binding to intracellular structures might cause a drastic reduction of availability of intracellular bioactive drug, reducing its effectivity [56,92]. Another possibility is that some antibiotics that reached the intracellular space get transported outside the cell again by means of p-glycoprotein efflux pumps [139] or by exocytosis [140]. However, it seems that granulocytes have a low expression of p-glycoprotein pumps [141]. When inside the cell, the drug also needs to reach the subcellular compartment, where *S. aureus* is residing. Inactivation of the antibiotic might occur inside the neutrophil. Some drug molecules are unstable under low pH conditions inside the phagosome, subsequently losing their antibiotic effectivity [75,86]. One or more of the last mechanisms might be the case for macrolide antibiotics like azithromycin, since the intracellular killing seems to be less than expected considering the high C/E ratio [86]. Lastly, although the effect of the antibiotics on neutrophil function did not seem fully responsible for intracellular antibiotic function or the lack thereof, this might be an interesting topic for further research. It could be investigated whether some drug molecules can be designed to boost neutrophil antibacterial functions in addition to their intrinsic antibacterial effect to enhance efficacy and antibacterial clearance.

### 3.2. Strengths and Limitations

Notably, the most frequently used parameter for cellular penetration of antibiotics, C/E ratios, varied greatly between the included studies. This is partly due to differences in experimental setup such as different incubation times. This creates difficulties comparing C/E ratios of the same drug extracted from different studies, where different incubation times have been used. An alternative parameter might be the intracellular AUC (time zero to infinity), which estimates the total intracellular drug exposure over time and not just at one single timepoint [55,61,66,79,121]. This diminishes the effect of variability of incubation time. Unfortunately, few in vivo studies could be included in this review, which makes it more difficult to make statements about clinical effectiveness in specific patient categories or conditions. Additionally, numerous studies showed difficulties discriminating between the direct intracellular antibacterial effects of antibiotics and the synergistic effects of antibiotics in combination with neutrophil killing [36,83]. Although it would be scientifically interesting to understand these underlying mechanisms, for clinical purposes this discrimination may be less important. Another point of discussion is the difference between techniques used for extracellular bacterial removal—differential centrifugation or addition of lysostaphin. As lysostaphin is able to penetrate neutrophils to some degree, addition of this agent to eliminate extracellular bacteria can result in overestimation of intracellular killing capacity [36]. The usage of different neutrophilic stimuli (i.e., PMA, zymosan, *S. aureus*) influences the penetration rate of the antibiotics [76,125]. In this review, neutrophils are the cells of interest. Caution is advised with extrapolation of the results to other cell types. Every cell type exerts a different cellular composition, which may result in differences in intracellular uptake, accumulation and action of the antibiotics [69].

### 3.3. Clinical Implications

Based on this review, rifamycins, quinolones and sulfamethoxazole-trimethoprim would be the preferred antibiotics to fight *S. aureus* infections in the presence of inadequate functioning neutrophils. Second choices would be macrolides and lincosamides and oxazolidinones. However, it is also important to consider other characteristics of these antibiotics and resistance patterns among *S. aureus*. Some of the most effective intracellular antibiotics, mainly the rifamycins, have some drawbacks regarding broad clinical use. Rifampin is known for its extensive drug interaction profile, for which monitoring is advised [142]. In addition, rifampin may cause some noteworthy adverse hematologic effects (leukopenia, anemia, thrombocytopenia) [143] and hepatotoxicity, mainly in patients at risk, like patients who use other potentially hepatotoxic drugs [144]. Linezolid is associated with reversible myelosuppression, of which a clinician should be aware and close monitoring is warranted [145,146]. Lincosamide clindamycin and quinolones are associated with a higher incidence of *Clostridium difficile* infections [147]. Macrolides and sulfamethoxazole-trimethoprim seem generally well tolerated in most patient groups, although with sulfamethoxazole-trimethoprim, caution is advised in immunocompromised (HIV) patients [148,149,150].

Given the rising threat of antibiotic resistance, it is important to emphasize the effectivity of intracellular effective antibiotics against resistant strains of *S. aureus*. In the EU/EEA population, the weighted mean MRSA percentage was 13.7% between 2013 and 2016. Among the MRSA population, combined resistance is common. For example, combination of MRSA with resistance to fluoroquinolones was seen in 10.7% of the population. Rifampin co-resistance is less common. Single resistance to fluoroquinolones was 5.3% and to rifampin, 0.4% [151]. However, (cross-)resistance to rifamycins is known to develop rapidly during monotherapy or high bacterial load [152]. MRSA is still largely susceptible to oxazolidinone linezolid and to sulfamethoxazole-trimethoprim, but some resistance is emerging [153,154]. Macrolide, lincosamide and streptogramin (MLS) cross-resistance is seen in *S. aureus*, mainly in MRSA, after the widespread use of macrolide compounds [155,156,157]. Based on the resistance patterns, rifampicin is unfavorable as monotherapy, linezolid and sulfamethoxazole-trimethoprim seem most favorable.

The current standard of care for methicillin susceptible *Staphylococcus* aureus (MSSA) bacteremia is mainly based on penicillins [158]. Vancomycin or cephalosporins, like cefazolin, are also used [159]. It is noteworthy that these antibiotics do not show convincing intracellular activity, as seen in this review. For MRSA, bacteremia glycopeptides are frequently used, and daptomycin also belongs to the treatment options. Glycopeptides and daptomycin also show insufficient intracellular effect. Cephalosporines, oxazolidinones (linezolid) or sulfamethoxazole-trimethoprim are used occasionally [159,160]. In some severe cases, although associated with some disadvantages, combination therapy is considered [161,162]. Combinations can be vancomycin/rifampin or vancomycin/gentamicin or daptomycin combined with gentamicin, rifampin, linezolid or sulfamethoxazole-trimethoprim [162]. Considering the current treatment standard, more focus should be placed on intracellular action of the frequently used antibiotics in bacteremia settings, especially when patients are susceptible to intracellular neutrophil infections, like is the case with immunocompromised and, as recently demonstrated, trauma patients [9,163]. Lack of intracellular effect can lead to a relapse of infection and treatment failure, even if the bacterial culture tests show broad antibiotic sensitivity. Since a substantial part of the frequently used antibiotics only reaches relatively low concentrations (sub-MIC) inside the cell, this contributes to the selection of bacteria and induction of resistance [164,165]. In this era of alarming antibiotic resistance, it is especially important to avoid this type of suboptimal antibiotic therapy and to focus on killing intracellular pathogens also.

## 4. Methods

### 4.1. Literature Search and Study Selection

A search was conducted using a broad search string (Appendix A). Searches were performed in MEDLINE and Embase databases in June 2017 and updated in August 2018. S.B. was mainly responsible for the search and study selection. Co-authors F.H. and P.H. were consulted in doubt. Disagreement was resolved by discussion. Only studies with primary data were included. The language had to be English or Dutch. No restrictions on publication date were applied. A summary of in- and exclusion criteria is displayed in Table 2. The articles found in the online databases were imported into reference manager Mendeley and the systematic review application Rayyan^®^. Duplicates were identified and removed. Subsequently, Rayyan^®^ was used to mark articles during the first screening process based on title and abstract, after which full text articles were imported into reference manager Mendeley and screened for applicability based on predetermined in- and exclusion criteria (Table 2). Articles without available abstracts and/or without full text were excluded. Inclusion and exclusion criteria in Table 2 are mostly based on the domain, determinant and outcome resulting from the review questions described in the introduction (Domain: Intracellular infection model with *S. aureus* in human neutrophils. Determinant: Clinically relevant antibiotics that can be systemically administered to humans. Outcome: Intracellular killing of *S. aureus* or factors influencing this process). 

### 4.2. Outcome Measures and Data Extraction

The focus was on the following outcome measures: Entry and accumulation of antibiotics inside neutrophils, their intracellular pharmacokinetics and distribution, intracellular activity of antibiotics and effects on intrinsic neutrophil functions (e.g., phagocytosis, reactive oxygen species (ROS) production, intracellular killing). All aforementioned outcome measures are relevant because they influence the final effect of an antibiotic on the killing of intracellular bacteria. Data regarding these outcome measures were extracted from the included articles by S.B. Co-authors F.H. and P.H. were consulted in doubt. Disagreement was resolved by discussion.

## 5. Conclusions

Quinolones, rifamycins and sulfamethoxazole-trimethoprim show the highest intracellular efficacy. Rifamycins are highly effective inside neutrophils but should be used with caution, considering their drug interaction profile, adverse effects and induction of resistance during monotherapy. Quinolones and sulfamethoxazole-trimethoprim are also very effective and should be drugs of first choice. This advice is applicable within the context of prophylactic therapy, for example, in severely injured trauma patients or as treatment in complicated infections suspected for *S. aureus* residing inside neutrophils. Oxazolidinones, macrolides and lincosamides also show good efficacy and can be considered drugs of second choice, regarding their intracellular actions against *S. aureus* in neutrophils.

## Figures and Tables

**Table 1 antibiotics-08-00054-t001:** Antibiotic penetration of the neutrophil, effect on neutrophil function and the effect of the antibiotics on intracellular *Staphylococcus aureus* at clinically relevant extracellular concentrations.

Class	Frequently Used Antibiotic	MIC mg/L [27]	Penetration (C/E Ratio)	Intra-Cellular Location	Effect on Neutrophil Function	Intracellular Effect on *S. aureus*	Type of Effect	Refences
Aminoglycosides	Gentamicin	0.125–2	<1	Low penetration. Lysosome	No effect, cytotoxic in high doses	Low	unknown	[28,29,30,31,32,33,34,35,36,37,38,39,40,41,42,43,44,45]
Tetracyclines ^§^	Tetracycline, doxycycline	0.125–1, 0.032–0.5	1.8–7.1 *	NS	No consensus. Contradicting results on ROS production	Moderate	bacteriostatic	[46,47,48,49,50,51]
Macrolides	Erythromycin, Azithromycin, Clarithromycin	0.064–1, 0.25–2, 0.064–0.5	4.4–34 and >100	Granules	No consensus. Contradicting results on ROS production	High	bacteriostatic	[29,30,32,36,38,40,41,47,52,53,54,55,56,57,58,59,60,61,62,63,64,65,66,67,68,69,70,71,72,73,74,75,76,77,78,79,80,81,82,83,84,85,86,87,88,89,90]
Lincosamides	Clindamycin	0.032–0.25	<3 and 8–43.4	Lysosome, cytosol	Induction phagocytosis. Contradicting results on ROS production	High	bacteriostatic	[29,30,31,34,36,38,40,42,43,45,59,61,78,83,91,92,93,94,95,96,97,98,99]
Oxazolidinones ^§^	Linezolid	0.5–4	<1 and 11	Lysosome, cytosol	Unknown	High	unknown	[37,100]
Penicillins	Benzylpenicillin, Amoxicillin	0.008–0.125, NR	<1	NS	No effect	Low/moderate	bacteriostatic (low dose); bactericidal (high dose)	[29,30,31,32,33,34,35,36,38,40,42,43,45,46,55,59,61,83,93,95,98,101,102,103,104,105,106,107,108,109,110]
Glycopeptides	Vancomycin	0.25–2	4–7.8 *	Cytosol	No effect	Low	unknown	[31,36,37,45,92,95,106,111,112,113]
Carbapenems ^§^	Meropenem	0.016–0.5	1–5	Cytosol	No effect	Low	unknown	[53,61,95,114]
Cephalosporins	Cefazolin, Ceftriaxone	0.125–2, 1–8	<1	Low penetration, phagosome	No effect	Low	unknown	[29,32,35,38,42,43,53,88,93,98,103,115,116,117,118]
Sulfonamides ^§^	Sulfamethoxazole	8–128	1.7–3.6	(Phago)lysosome, cytosol	Unknown	#	unknown	[42,46,119,120]
DHFR Inhibitors ^§^	Trimethoprim	0.25–2	3–21*	Cytosol, microsomal	Dose-dependent inhibition ROS-production	#	unknown	[30,42,43,46,53,61,93,120,121]
Quinolones	Ciprofloxacin	0.064–1	2.2–10.9	NS	No effect	Very high	bacteriostatic (low dose); bactericidal (high dose)	[31,36,45,84,88,93,95,104,105,107,122,123,124,125,126,127,128,129,130,131,132]
Rifamycins	Rifampin	0.004–0.032	2.3–9.8 *	Phagosome	No effect	Very high	bactericidal	[29,31,33,34,35,36,37,38,39,40,42,43,44,45,52,83,92,93,94,101,108,110,113,119,133,134]
Others	Fosfomycin ^§^	0.25–32	1.8	NS	Enhanced phagocytosis, ROS production, NETosis	Low/moderate	bactericidal	[113,119,135]
Others	Daptomycin ^§^	0.064–1	<1	NS	No effect	Low	unknown	[31,136]
Others	Sulfamethoxazole/Trimethoprim	0.032–0.5	&	NS	No effect	Very high	unknown	[34,37,42,43,46,119]
Others	Amoxicillin/Clavulanic acid ^§^	NR	&	NS	No effect at a clinically relevant level	Low	unknown	[137]

Abbreviations: NR not reported; NS not specified. * Outliers are not taken into account; # See Sulfamethoxazole-trimethoprim combination; & See individual antibiotics; ^§^ Very little evidence.

**Table 2 antibiotics-08-00054-t002:** In- and exclusion criteria for study selection.

	Inclusion Criteria	Exclusion Criteria
1	Intracellular *S. aureus* infection model or information that can be extrapolated to this situation	Infection model with other micro-organism than *S. aureus* or extracellular infection model
2	Cell type is human neutrophils	Non-human cells or other cells than neutrophils
3	Data regarding primary endpoints	Data regarding fundamental pathophysiological mechanisms, a specific method or technique or novel drug delivery methods
4	Clinically relevant antibiotic(s), administered systemically in humans	Antibiotics only used in experimental setting, non-registered antibiotics for human use, antibiotics not frequently used in the clinic * or non-systemic antibiotics
5	Normal functioning neutrophils, not under influence of a systemic disease influencing neutrophils **	Conditions or systemic diseases not of interest playing a primary role in the study design (i.e., cancer or auto-immune conditions)

* Some data regarding experimental or not frequently used antibiotics is still included if it is relevant to illustrate properties of a whole class of antibiotics. ** Some data concerning neutrophils of immunocompromised patients (i.e., chronic granulomatous disease (CGD), AIDS, diabetes) were considered relevant for the scope of this review.

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
