# Peer review of "Intracellular Penetration and Effects of Antibiotics on Staphylococcus aureus Inside Human Neutrophils: A Comprehensive Review"

_antibiotics, 2019, doi:10.3390/antibiotics8020054_

Round 1

Reviewer 1 Report

Intracellular penetration and effects of antibiotics on S. aureus inside human neutrophils: a comprehensive review

Bongers et al

Bongers et al have conducted a valuable survey of literature that discuss effects of clinical antibiotics on intracellular S aureus infections, and their neutrophil hosts. The content is organized based on several antibiotic classes and collects information from many examples. Overall, this is an important review that will be of use to the readership of Antibiotics. Consideration of the following issues would help to improve the paper.

1. An overview of neutrophils in the introduction would be very helpful to understanding some of the obervations about abx localization, etc. It would help provide a context for the rest of the review. It is likely readership is familiar with the mechanism of action of the abx, but may not be as familiar with neutrophil function.

2. Table 1. It would be helpful to add a column for MIC/MBC for abx, perhaps susceptible and resistant? The “effect on neutrophil” column should have more detail in it; this is in conjuction with comment #1. If the neutrophil structure and function are clearer, more info can be put into this column of the table.

3. Neutrophil entry. C/E ratio is pretty clear (authors should be consistent with . or , to separate units), but they should have a consistent number of significant figures. The point will still be made. The text also puts emphasis on kinetics/rate of entry but the significance is not clear, this point could be brought out a little more. What are structural features on abx that facilitate transport?

4. Could Table 1 also have a column that contains the speculation of the authors about the probability of synergistic effects (activate neutrophils, kill Sa) of each entry? This could also be discussed a little more in the discussion. This review has a chance to guide researchers toward properties that should be incorporated into new antibiotics.

5. There are several unclear statements that should be addressed, in addition to a few typos, English usage issues.

What are non-professional phagocytes??p2 line 46..

            What is,” This effect was partly but not  completely due to regurgitation (and consecutive extracellular killing)) or procedural imperfections.“?

            Page 2 paragraphs 2 and 3 are hard to understand.

Author Response

Bongers et al have conducted a valuable survey of literature that discuss effects of clinical antibiotics on intracellular S aureus infections, and their neutrophil hosts. The content is organized based on several antibiotic classes and collects information from many examples. Overall, this is an important review that will be of use to the readership of Antibiotics. Consideration of the following issues would help to improve the paper.

1. An overview of neutrophils in the introduction would be very helpful to understanding some of the obervations about abx localization, etc. It would help provide a context for the rest of the review. It is likely readership is familiar with the mechanism of action of the abx, but may not be as familiar with neutrophil function.

-       This is a very relevant remark, indeed for better understanding of the review some basic principles on bacterial killing by neutrophils is important. We added a few lines to the beginning of the introduction to clarify the basics of bacterial killing. A process involving mainly phagocytosis, fusion of the phagosome with neutrophil granules and vesicles, and subsequent acidification on proteases activity. (Page 1, Line 31-37)

2. Table 1. It would be helpful to add a column for MIC/MBC for abx, perhaps susceptible and resistant? The “effect on neutrophil” column should have more detail in it; this is in conjuction with comment #1. If the neutrophil structure and function are clearer, more info can be put into this column of the table.

-       Again very valuable additions to the main table. We feel that the suggested changes (localization within the neutrophils, MIC , clarification of headings) enables us to get the message across more clearly. Together with the addition of the references (suggested by reviewer 3), table 1 is now a complete overview of the literature.  (Page 2/3, table 1)

3. Neutrophil entry. C/E ratio is pretty clear (authors should be consistent with . or , to separate units but they should have a consistent number of significant figures. The point will still be made. The text also puts emphasis on kinetics/rate of entry but the significance is not clear, this point could be brought out a little more. What are structural features on abx that facilitate transport?

-       Many thanks for the critical remarks. We regret some inconsistencies were left in the text. We made the suggested changes in table on were “,” was changed to “.”, in line with the main body of text. All C/E ratios were altered to one decimal point in order to maintain consistency. In regard of the kinetic characteristics of intracellular antibiotics, a few sentences have been added to point out beneficial characteristics of intracellular antibiotics (lines 437-439, 441-444)

4. Could Table 1 also have a column that contains the speculation of the authors about the probability of synergistic effects (activate neutrophils, kill Sa) of each entry? This could also be discussed a little more in the discussion. This review has a chance to guide researchers toward properties that should be incorporated into new antibiotics.

-       In order to give suggestions for better intracellular antibiotics, a few lines have been added to the text, pointing out beneficial characteristics of intracellular active antibiotics (see also the comment above). Lines 437-439, 441-444, 456-459. For this no extra column is added to the table in order to reduce the size of the table.

5. There are several unclear statements that should be addressed, in addition to a few typos, English usage issues.

What are non-professional phagocytes??p2 line 46

-       To clarify this matter the examples from the cited references are added to our text (page 2, line 54).  

            What is,” This effect was partly but not  completely due to regurgitation (and consecutive extracellular killing)) or procedural imperfections.“?

-       For clarification, the sentence was rephrased: “This effect was partly but not completely due to overestimation of the amount of killed intracellular bacteria [43].” Page 4 (line107-108)

            Page 2 paragraphs 2 and 3 are hard to understand

-       We regret that these paragraphs were hard to understand. Thank you very much for the feedback. We changed sentence structure to make in more readable. We hope you agree, if not, of course we are more than willing to make some additional changes. (page 2 line 41-62)  

Reviewer 2 Report

Overall a nicely comprehensive review of the field. Minor comments:

S. aureus throughout is not italicised;

lines 117-120: inconsistent use of significant figures;

line 126: telithomycin is missing an "r"

line 129: extent, not extend

line 434: commentary is given to suggest that measuring intracellular AUC would be a better predictor of cellular penetration, to compensate for differing methodology that potentially complicates direct comparison of C/E ratios between different studies.

How does methodology differ between measuring C/E ratios vs total AUC ? can the authors comment on the differences and how the latter would provide an advantage from a technical point of view?

line 320: Lastly, with reference to the cidal antibiotic classes reviewed (quinolones, lactams and aminoglycosides), the quinolones appear to penetrate neutrophils the most effectively. The 3 classes are also controversially implicated as being involved in downstream ROS production as part of the killing mechanism. How does this phenomenon influence ROS production/mitochondrial dysfunction in neutrophils during antibiotic accumulation? Is anything known about this?

Author Response

S. aureus throughout is not italicised;

-       Thank you for this comment. We made the suggested changes throughout the manuscript.

lines 117-120: inconsistent use of significant figures;

-       Many thanks for the critical remarks. All C/E ratios were altered throughout the manuscript to one decimal point in order to maintain consistency.

line 126: telithomycin is missing an "r" . à Thank you. The suggested changes are made.

line 129: extent, not extend à Thank you. The suggested changes are made.

line 434: commentary is given to suggest that measuring intracellular AUC would be a better predictor of cellular penetration, to compensate for differing methodology that potentially complicates direct comparison of C/E ratios between different studies. How does methodology differ between measuring C/E ratios vs total AUC ? can the authors comment on the differences and how the latter would provide an advantage from a technical point of view?

-       This is indeed a very relevant point. In our discussion, we feel we addressed this issue in the subsection ‘strengths and limitations’. As C/E ratios is the most commonly used parameter in the included studies, we could not review otherwise. However, as suggested, AUC could be an interesting alternative given the fact that the latter is less influenced by timing of the experimental and measurements. We adjusted this part of the text in order to give a clearer explanation about why AUC might be a good alternative parameter. Lines 465-467. If the reviewers wants us to address this matter in further detail, we are more than willing to do so.

line 320: Lastly, with reference to the cidal antibiotic classes reviewed (quinolones, lactams and aminoglycosides), the quinolones appear to penetrate neutrophils the most effectively. The 3 classes are also controversially implicated as being involved in downstream ROS production as part of the killing mechanism. How does this phenomenon influence ROS production/mitochondrial dysfunction in neutrophils during antibiotic accumulation? Is anything known about this?

-       Thank you for the interesting comment. Just as the reviewer, we were very curious to know which mechanisms are involved. Unfortunately, this is neither addressed in studies at hand, nor in other relevant literature. A lot of relevant research can be done in neutrophil pathways.

Reviewer 3 Report

The overall idea of the manuscript is valid, the subject is relevant in the antibiotics field and the strategy of the manuscript is well established and makes sense. The manuscript is well written and the English language correct. However, there are some aspects in which the manuscript should be ameliorated. If the authors agree to do these improvements, I recommend the acceptance of the manuscript to be published in Antibiotics with minor revision:

- The methodology used for the bibliographic search should appear before the results;

- The English language in Table 1 caption needs to be improved. Please revise the text;

- In table one authors should add a column with the references used to each antibiotics class;

- Better explain what is the meaning of “effect on neutrophil” and “intracellular effect” in table 1; to what kind of effects they refer to? The information should be more precise even when a consensus is not reached in the cited bibliography;

- The meaning of C/E ratio should be explained before table 1 or at the end of this table; also, the calculation of this ration should be also explained, the same unit are compared (molar concentration, mass concentration?);

- The neutrophils referred along the text are freshly isolated human neutrophils in all the cited studies? What are the times of exposure to the antibiotics? What the used concentrations? This information should be clearly stated for all the cited studies. Authors should clarify these points due to their biological significance;

- The cited in vivo studies should have a reference to the in vivo model used; the administered dosses are comparable to the ones administered to humans?

- Improve the sentence: “Clindamycin slightly induces phagocytosis by neutrophils at serum…”;

- Clarify and complete this statement in what respect bacterial characteristics: “This might be due to modification of bacterial characteristics [36].”;

- Correct to formal English: “…acid don’t show…”.

Author Response

The overall idea of the manuscript is valid, the subject is relevant in the antibiotics field and the strategy of the manuscript is well established and makes sense. The manuscript is well written and the English language correct. However, there are some aspects in which the manuscript should be ameliorated. If the authors agree to do these improvements, I recommend the acceptance of the manuscript to be published in Antibiotics with minor revision:

The methodology used for the bibliographic search should appear before the results;

-       Although this might be the conventional way in some journals. In Antibiotics the format is to show the ‘Methods’ section after ‘Results’ and ‘Discussion’. So unfortunately, the changes could not be made.

-The English language in Table 1 caption needs to be improved. Please revise the text;

-       Gratitude for the relevant remark, we agree that there could be improvement to the caption. We adjusted it to:

Antibiotic penetration of the neutrophil, effect on neutrophil function and the effect of the antibiotics on intracellular S. aureus at clinically relevant extracellular concentrations. (page 2/3, table 1)

We feel that with the alterations the caption is more readable and in correct English. We hope the reviewer agrees.

In table one authors should add a column with the references used to each antibiotics class;

-       This is a very important comment and we totally agree this should be displayed in the table. For this we added a column with references. (page 2/3 table 1)

Better explain what is the meaning of “effect on neutrophil” and “intracellular effect” in table 1; to what kind of effects they refer to? The information should be more precise even when a consensus is not reached in the cited bibliography;

-       Again very valuable additions to the main table. We feel that the suggested changes (clarification of headings, and effect of antiobtiocs) enables us to get the message across more clearly. Together with the addition of the references (suggested above) table 1 is now a more complete and precise overview of the literature.  (Page 2/3, table 1). We also clarified the description of table 1 in the main text (lines 77-79)

The meaning of C/E ratio should be explained before table 1 or at the end of this table; also, the calculation of this ration should be also explained, the same unit are compared (molar concentration, mass concentration?);

-       It is indeed important to explain this ratio before mentioning it in table 1. Therefore we added an explanation in the text above table 1 (lines 76-79)

The neutrophils referred along the text are freshly isolated human neutrophils in all the cited studies? What are the times of exposure to the antibiotics? What the used concentrations? This information should be clearly stated for all the cited studies. Authors should clarify these points due to their biological significance;

-       Although we agree with the reviewer that the asked information is of great importance to be able to interpret the results. However, we feel that most relevant information on incubation time and time points are already stated in the text. At least this is true for the relevant results stated in our review. So however relevant, we feel we can not described every experimental setup detail for readability of the manuscript. We hope that the reviewer can understand our standpoint.

The cited in vivo studies should have a reference to the in vivo model used; the administered dosses are comparable to the ones administered to humans?

-       This question is in line with the previous comment. In case of in vivo studies it is stated if the doses used in the experiments were clinically ‘relevant’ or not. In case of clinical relevant doses, it concerned the dose regularly administered in human for therapeutic purposes.

Improve the sentence: “Clindamycin slightly induces phagocytosis by neutrophils at serum…”;

-       Thanks for the suggestion. The sentences is corrected to “At serum concentrations of 1.0 mg/L clindamycin slightly enhances neutrophil phagocytosis  in healthy volunteers” (page 6, line 212). We feel that the changes made are an improvement and hope the reviewer agrees.

Clarify and complete this statement in what respect bacterial characteristics: “This might be due to modification of bacterial characteristics [36].”;

-       We regret the indistinctness, and the needed alterations for clarifications are made: “This might be due to modification of bacterial characteristics by modification of opsonization or interaction between the antibiotic and the bacterial inoculum in vitro before ingestion” (lines 282-283).  Again, many thanks for the valuable remarks. 

Correct to formal English: “…acid don’t show…”.

-       Thank you. The suggested changes are made (page 11, line 451)